# Robustness May be at Odds with Accuracy

**Dimitris Tsipras,**[*] **Shibani Santurkar,**[*] **Logan Engstrom,**[*] **Alexander Turner, Aleksander Mądry**
Massachusetts Institute of Technology
{tsipras,shibani,engstrom,turneram,madry}@mit.edu

## Abstract

We show that there exists an inherent tension between the goal of adversarial robustness and that of standard generalization. Specifically, training robust models may not only be more resource-consuming, but also lead to a reduction of standard accuracy. We demonstrate that this trade-off between the standard accuracy of a model and its robustness to adversarial perturbations *provably* exists even in a fairly simple and natural setting. These findings also corroborate a similar phenomenon observed in practice. Further, we argue that this phenomenon is a consequence of robust classifiers learning fundamentally different feature representations than standard classifiers. These differences, in particular, seem to result in unexpected benefits: the features learned by robust models tend to align better with salient data characteristics and human perception.

## 1 Introduction

Deep learning models have achieved impressive performance on a number of challenging benchmarks in computer vision, speech recognition and competitive game playing (Krizhevsky et al., 2012; Graves et al., 2013; Mnih et al., 2015; Silver et al., 2016; He et al., 2015a). However, it turns out that these models are actually quite brittle. In particular, one can often synthesize small, imperceptible perturbations of the input data and cause the model to make highly-confident but erroneous predictions (Dalvi et al., 2004; Biggio & Roli, 2017; Szegedy et al., 2013).

This problem of so-called *adversarial examples* has garnered significant attention recently and resulted in a number of approaches both to finding these perturbations, and to training models that are robust to them (Goodfellow et al., 2014b; Nguyen et al., 2015; Moosavi-Dezfooli et al., 2016; Carlini & Wagner, 2016; Sharif et al., 2016; Kurakin et al., 2016a; Evtimov et al., 2017; Athalye et al., 2017). However, building such *adversarially robust* models has proved to be quite challenging. In particular, many of the proposed robust training methods were subsequently shown to be ineffective (Carlini & Wagner, 2017; Athalye et al., 2018; Uesato et al., 2018). Only recently, has there been progress towards models that achieve robustness that can be demonstrated empirically and, in some cases, even formally verified (Madry et al., 2017; Kolter & Wong, 2017; Sinha et al., 2017; Tjeng & Tedrake, 2017; Raghunathan et al., 2018; Dvijotham et al., 2018a; Xiao et al., 2018b).

The vulnerability of models trained using standard methods to adversarial perturbations makes it clear that the paradigm of adversarially robust learning is different from the classic learning setting. In particular, we already know that robustness comes at a cost. This cost takes the form of computationally expensive training methods (more training time), but also, as shown recently in Schmidt et al. (2018), the potential need for more training data. It is natural then to wonder: *Are these the only costs of adversarial robustness?* And, if so, once we choose to pay these costs, *would it always be preferable to have a robust model instead of a standard one?* The goal of this work is to explore these questions and thus, in turn, to bring us closer to understanding the phenomenon of adversarial robustness.

**Our contributions**   It might be natural to expect that training models to be adversarially robust, albeit more resource-consuming, can only improve performance in the standard classification setting. In this work, we show, however, that the picture here is much more nuanced: these two goals might be fundamentally at odds. Specifically, even though applying adversarial training, the leading method

---

[*]Equal Contribution.

for training robust models, can be beneficial in some regimes of training data size, in general, there is a trade-off between the *standard accuracy* and *adversarially robust accuracy* of a model. In fact, we show that this trade-off *provably* exists even in a fairly simple and natural setting.

At the root of this trade-off is the fact that features learned by the optimal standard and optimal robust classifiers are fundamentally different and, interestingly, this phenomenon persists even in the limit of infinite data. This thus also goes against the natural expectation that given sufficient data, classic machine learning tools would be sufficient to learn robust models and emphasizes the need for techniques specifically tailored to training robust models.

Our exploration also uncovers certain unexpected benefit of adversarially robust models. In particular, adversarially robust learning tends to equip the resulting models with invariances that we would expect to be also present in human vision. This, in turn, leads to features that align better with human perception, and could also pave the way towards building models that are easier to understand. Consequently, the feature embeddings learnt by robust models yield also clean inter-class interpolations, similar to those found by generative adversarial networks (GANs) (Goodfellow et al., 2014b) and other generative models. This hints at the existence of a stronger connection between GANs and adversarial robustness.

## 2 ON THE PRICE OF ADVERSARIAL ROBUSTNESS

Recall that in the canonical classification setting, the primary focus is on maximizing standard accuracy, i.e. the performance on (yet) unseen samples from the underlying distribution. Specifically, the goal is to train models that have low *expected loss* (also known as population risk):

$$\mathbb{E}_{(x,y)\sim\mathcal{D}}[\mathcal{L}(x,y;\theta)]. \tag{1}$$

**Adversarial robustness** The existence of adversarial examples largely changed this picture. In particular, there has been a lot of interest in developing models that are resistant to them, or, in other words, models that are *adversarially robust*. In this context, the goal is to train models that have low *expected adversarial loss*:

$$\mathbb{E}_{(x,y)\sim\mathcal{D}}\left[\max_{\delta\in\Delta}\mathcal{L}(x+\delta,y;\theta)\right]. \tag{2}$$

Here, $\Delta$ represents the set of perturbations that the adversary can apply to induce misclassification. In this work, we focus on the case when $\Delta$ is the set of $\ell_p$-bounded perturbations, i.e. $\Delta = \{\delta \in \mathbb{R}^d \mid \|\delta\|_p \leq \varepsilon\}$. This choice is the most common one in the context of adversarial examples and serves as a standard benchmark. It is worth noting though that several other notions of adversarial perturbations have been studied. These include rotations and translations (Fawzi & Frossard, 2015; Engstrom et al., 2017), and smooth spatial deformations (Xiao et al., 2018a). In general, determining the "right" $\Delta$ to use is a domain specific question.

**Adversarial training** The most successful approach to building adversarially robust models so far (Madry et al., 2017; Kolter & Wong, 2017; Sinha et al., 2017; Raghunathan et al., 2018) was so-called *adversarial training* (Goodfellow et al., 2014b). Adversarial training is motivated by viewing (2) as a statistical learning question, for which we need to solve the corresponding (adversarial) empirical risk minimization problem:

$$\min_{\theta} \mathbb{E}_{(x,y)\sim\widehat{\mathcal{D}}}\left[\max_{\delta\in S}\mathcal{L}(x+\delta,y;\theta)\right].$$

The resulting saddle point problem can be hard to solve in general. However, it turns out to be often tractable in practice, at least in the context of $\ell_p$-bounded perturbations (Madry et al., 2017). Specifically, adversarial training corresponds to a natural robust optimization approach to solving this problem (Ben-Tal et al., 2009). In this approach, we repeatedly find the *worst-case* input perturbations $\delta$ (solving the inner maximization problem), and then update the model parameters to reduce the loss on these perturbed inputs.

Though adversarial training is effective, this success comes with certain drawbacks. The most obvious one is an increase in the training time (we need to compute new perturbations each parameter update

step). Another one is the potential need for more training data as shown recently in (Schmidt et al., 2018). These costs make training more demanding, but is that the whole price of being adversarially robust? In particular, if we are willing to pay these costs: *Are robust classifiers better than standard ones in every other aspect?* This is the key question that motivates our work.

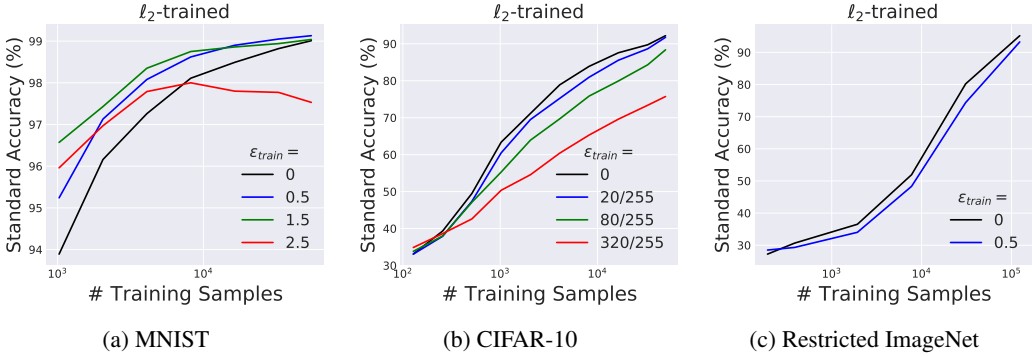

(a) MNIST        (b) CIFAR-10        (c) Restricted ImageNet

Figure 1: Comparison of the standard accuracy of models trained against an $\ell_2$-bounded adversary as a function of size of the training dataset. We observe that when training with few samples, adversarial training has a positive effect on model generalization (especially on MNIST). However, as training data increase, the standard accuracy of robust models drops below that of the standard model ($\varepsilon_{train} = 0$). Similar results for $\ell_\infty$ trained networks are shown in Figure 6 of Appendix G.

**Adversarial Training as a Form of Data Augmentation**   Our point of start is a popular view of adversarial training as the "ultimate" form of data augmentation. According to this view, the adversarial perturbation set $\Delta$ is seen as the set of invariants that a good model should satisfy (regardless of the adversarial robustness considerations). Thus, finding the worst-case $\delta$ corresponds to augmenting the training data in the "most confusing" and thus also "most helpful" manner. A key implication of this view is that adversarial training should be beneficial for the standard accuracy of a model (Torkamani & Lowd, 2013; 2014; Goodfellow et al., 2014b; Miyato et al., 2018).

Indeed, in Figure 1, we see this effect, when classifiers are trained with relatively few samples (particularly on MNIST). In this setting, the amount of training data available is insufficient to learn a good standard classifier and the set of adversarial perturbations used is "compatible" with the learning task. (That is, good *standard* models for this task need to be also somewhat invariant to these perturbations.) In such regime, robust training does indeed act as data augmentation, regularizing the model and leading to a better solution (from standard accuracy point of view). (Note that this effect seems less pronounced for CIFAR-10, possibly because $\ell_p$-invariance is not as important for a good standard CIFAR-10 classifier.)

Surprisingly however, in Figure 6 we see that as we include more samples in the training set, this positive effect becomes less significant. In fact, after some point adversarial training actually *decreases* the standard accuracy. In Figure 7 in Appendix G we study the behaviour of models trained using adversarial training with different $\ell_p$-bounded adversaries. We observe a steady decline in standard accuracy as the strength of the adversary increases. (Note that this still holds if we train on batches that also contain natural examples, as in Kurakin et al. (2016a). See Appendix B.) Similar effects were also observed in prior work (Kurakin et al., 2016b; Madry et al., 2017; Dvijotham et al., 2018b; Wong et al., 2018; Xiao et al., 2018b; Su et al., 2018; Babbar & Schölkopf, 2018).

The goal of this work is to illustrate and explain the roots of this phenomenon. In particular, we would like to understand:

> *Why does there seem to be a trade-off between standard and adversarially robust accuracy?*

As we will show, this effect is not an artifact of our adversarial training methods but in fact is inevitable consequence of different goals of adversarial robustness and standard generalization.

## 2.1 Adversarial robustness might be incompatible with standard accuracy

As we discussed above, we often observe that employing adversarial training leads to a decrease in a model's standard accuracy. In what follows, we show that this phenomenon is a manifestation of an inherent tension between standard accuracy and adversarially robust accuracy. In particular, we present a theoretical model that demonstrates it. In fact, this phenomenon can be illustrated in a fairly simple setting which suggests that it is quite prevalent.

**Our binary classification task** Our data model consists of input-label pairs $(x, y)$ sampled from a distribution $\mathcal{D}$ as follows:

$$y \overset{u.a.r}{\sim} \{-1, +1\}, \qquad x_1 = \begin{cases} +y, & \text{w.p. } p \\ -y, & \text{w.p. } 1-p \end{cases}, \qquad x_2, \ldots, x_{d+1} \overset{i.i.d}{\sim} \mathcal{N}(\eta y, 1), \qquad (3)$$

where $\mathcal{N}(\mu, \sigma^2)$ is a normal distribution with mean $\mu$ and variance $\sigma^2$, and $p \geq 0.5$. We chose $\eta$ to be large enough so that a simple classifier attains high standard accuracy (>99%) – e.g. $\eta = \Theta(1/\sqrt{d})$ will suffice. The parameter $p$ quantifies how correlated the feature $x_1$ is with the label. For the sake of example, we can think of $p$ as being $0.95$. This choice is fairly arbitrary; the trade-off between standard and robust accuracy will be qualitatively similar for any $p < 1$.

**Standard classification is easy** Note that samples from $\mathcal{D}$ consist of a single feature that is *moderately correlated* with the label and $d$ other features that are only *very weakly* correlated with it. Despite the fact that each one of the latter type of features individually is hardly predictive of the correct label, this distribution turns out to be fairly simple to classify from a standard accuracy perspective. Specifically, a natural (linear) classifier

$$f_{\text{avg}}(x) := \text{sign}(w_{\text{unif}}^{\top} x), \quad \text{where } w_{\text{unif}} := \left[0, \frac{1}{d}, \ldots, \frac{1}{d}\right], \qquad (4)$$

achieves standard accuracy arbitrarily close to $100\%$, for $d$ large enough. Indeed, observe that

$$\Pr[f_{\text{avg}}(x) = y] = \Pr[\text{sign}(w_{\text{unif}}x) = y] = \Pr\left[\frac{y}{d}\sum_{i=1}^{d}\mathcal{N}(\eta y, 1) > 0\right] = \Pr\left[\mathcal{N}\left(\eta, \frac{1}{d}\right) > 0\right],$$

which is $> 99\%$ when $\eta \geq 3/\sqrt{d}$.

**Adversarially robust classification** Note that in our discussion so far, we effectively viewed the average of $x_2, \ldots, x_{d+1}$ as a single "meta-feature" that is highly correlated with the correct label. For a standard classifier, any feature that is even slightly correlated with the label is useful. As a result, a standard classifier will take advantage (and thus rely on) the weakly correlated features $x_2, \ldots, x_{d+1}$ (by implicitly pooling information) to achieve almost perfect standard accuracy.

However, this analogy breaks completely in the adversarial setting. In particular, an $\ell_{\infty}$-bounded adversary that is only allowed to perturb each feature by a moderate $\varepsilon$ can effectively override the effect of the aforementioned meta-feature. For instance, if $\varepsilon = 2\eta$, an adversary can shift each weakly-correlated feature towards $-y$. The classifier would now see a perturbed input $x'$ such that each of the features $x'_2, \ldots, x'_{d+1}$ are sampled i.i.d. from $\mathcal{N}(-\eta y, 1)$ (i.e., now becoming *anti*-correlated with the correct label). Thus, when $\varepsilon \geq 2\eta$, the adversary can essentially simulate the distribution of the weakly-correlated features as if belonging to the wrong class.

Formally, the probability of the meta-feature correctly predicting $y$ in this setting (4) is

$$\min_{\|\delta\|_{\infty} \leq \varepsilon} \Pr[\text{sign}(x + \delta) = y] = \Pr[\mathcal{N}(\eta, 1) - \varepsilon > 0] = \Pr[\mathcal{N}(-\eta, 1) > 0].$$

As a result, the simple classifier in (4) that relies solely on these features cannot get adversarial accuracy better than $1\%$.

Intriguingly, this discussion draws a distinction between *robust* features ($x_1$) and *non-robust* features ($x_2, \ldots, x_{d+1}$) that arises in the adversarial setting. While the meta-feature is far more predictive of the true label, it is extremely unreliable in the presence of an adversary. Hence, a tension between standard and adversarial accuracy arises. Any classifier that aims for high accuracy (say $> 99\%$) will

have to heavily rely on non-robust features (the robust feature provides only, say, $95\%$ accuracy). However, since the non-robust features can be arbitrarily manipulated, this classifier will inevitably have low adversarial accuracy. We make this formal in the following theorem proved in Appendix C.

**Theorem 2.1** (Robustness-accuracy trade-off). *Any classifier that attains at least $1 - \delta$ standard accuracy on $\mathcal{D}$ has robust accuracy at most $\frac{p}{1-p}\delta$ against an $\ell_\infty$-bounded adversary with $\varepsilon \geq 2\eta$.*

This bound implies that if $p < 1$, as standard accuracy approaches $100\%$ ($\delta \to 0$), adversarial accuracy falls to $0\%$. As a concrete example, consider $p = 0.95$, then any classifier with standard accuracy more than $1 - \delta$ will have robust accuracy at most $19\delta$[1]. Also it is worth noting that the theorem is tight. If $\delta = 1 - p$, both the standard and adversarial accuracies are bounded by $p$ which is attained by the classifier that relies solely on the first feature. Additionally, note that compared to the scale of the features $\pm 1$, the value of $\varepsilon$ required to manipulate the standard classifier is very small ($\varepsilon = O(\eta)$, where $\eta = O(1/\sqrt{d})$).

**On the (non-)existence of an accurate and robust classifier** It might be natural to expect that in the regime of infinite data, the standard classifier itself acts as a robust classifier. Note however, that this is not true for the setting we analyze above. Here, the trade-off between standard and adversarial accuracy is an inherent trait of the data distribution itself and not due to having insufficient samples. In this particular classification task, we (implicitly) assumed that there does not exist a classifier that is *both* robust and very accurate (i.e. $> 99\%$ standard and robust accuracy). Thus, for this task, any classifier that is very accurate (including the Bayes classifier – the classifier minimizing classification error having full-information about the distribution) will necessarily be non-robust.

This seemingly goes against the common assumption in adversarial ML that humans are such perfect robust and accurate classifiers for standard datasets. However, note that there is no concrete evidence supporting this assumption. In fact, humans often have far from perfect performance in vision benchmarks (Karpathy, 2011; 2014; Russakovsky et al., 2015) and are outperformed by ML models in certain tasks (He et al., 2015b; Gastaldi, 2017). It is plausible that standard ML models are able to outperform humans in these tasks by relying on brittle features that humans are naturally invariant to and the observed decrease in performance might be the manifestation of that.

## 2.2 THE IMPORTANCE OF ADVERSARIAL TRAINING

As we have seen in the distributional model $\mathcal{D}$ (3), a classifier that achieves very high standard accuracy (1) will inevitably have near-zero adversarial accuracy. This is true even when a classifier with reasonable standard and robust accuracy exists. Hence, in an adversarial setting (2), where the goal is to achieve high adversarial accuracy, the training procedure needs to be modified. We now make this phenomenon concrete for linear classifiers trained using the soft-margin SVM loss. Specifically, in Appendix D we prove the following theorem.

**Theorem 2.2** (Adversarial training matters). *For $\eta \geq 4/\sqrt{d}$ and $p \leq 0.975$ (the first feature is not perfect), a soft-margin SVM classifier of unit weight norm minimizing the distributional loss achieves a standard accuracy of $> 99\%$ and adversarial accuracy of $< 1\%$ against an $\ell_\infty$-bounded adversary of $\varepsilon \geq 2\eta$. Minimizing the distributional adversarial loss instead leads to a robust classifier that has standard and adversarial accuracy of $p$ against any $\varepsilon < 1$.*

This theorem shows that if our focus is on robust models, adversarial training is crucial to achieve non-trivial adversarial accuracy in this setting. Simply optimizing the standard accuracy of the model (i.e. standard training) leads to poor robust accuracy. Soft-margin SVM classifiers and the constant $0.975$ are chosen for mathematical convenience. Our proofs do not depend on them in a crucial way and can be adapted, in a straightforward manner, to other natural settings, e.g. logistic regression.

**Transferability** An interesting implication of our analysis is that standard training produces classifiers that rely on features that are weakly correlated with the correct label. This will be true for any classifier trained on the same distribution. Hence, the adversarial examples that are created by perturbing each feature in the direction of $-y$ will transfer across classifiers trained on independent

---

[1]Hence, any classifier with standard accuracy $\geq 99\%$ has robust accuracy $\leq 19\%$ and any classifier with standard accuracy $\geq 96\%$ has robust accuracy $\leq 76\%$.

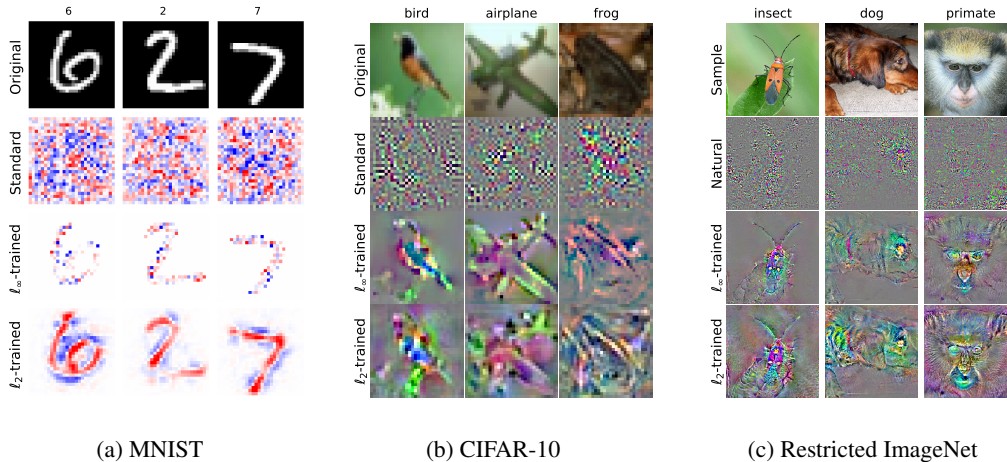

(a) MNIST        (b) CIFAR-10        (c) Restricted ImageNet

Figure 2: Visualization of the loss gradient with respect to input pixels. Recall that these gradients highlight the input features which affect the loss most strongly, and thus are important for the classifier's prediction. We observe that the gradients are significantly more *interpretable* for adversarially trained networks – they align well with perceptually relevant features. In contrast, for standard networks they appear very noisy. We observe that gradients of $\ell_\infty$-trained models tend to be sparser than those of $\ell_2$-trained models. (For MNIST, blue and red pixels denote positive and negative gradient regions respectively. For CIFAR-10 and ImageNet, we clip gradients to within $\pm 3\sigma$ and rescale them to lie in the $[0,1]$ range.) Additional visualizations are in Figure 10 of Appendix G.

samples from the distribution. This constitutes an interesting manifestation of the generally observed phenomenon of transferability (Szegedy et al., 2013) and might hint at its origin.

**Empirical examination** In Section 2.1, we showed that the trade-off between standard accuracy and robustness might be inevitable. To examine how representative our theoretical model is of real-world datasets, we also experimentally investigate this issue on MNIST (LeCun et al., 1998) as it is amenable to linear classifiers. Interestingly, we observe a qualitatively similar behavior. For instance, in Figure 5(b) in Appendix E, we see that the standard classifier assigns weight to even weakly-correlated features. (Note that in settings with finite training data, such brittle features could arise even from noise – see Appendix E.) The robust classifier on the other hand does not assign any weight beyond a certain threshold. Further, we find that it is possible to obtain a robust classifier by directly training a *standard* model using only features that are relatively well-correlated with the label (without adversarial training). As expected, as more features are incorporated into the training, the standard accuracy is improved at the cost of robustness (see Appendix E Figure 5(c)).

## 3 UNEXPECTED BENEFITS OF ADVERSARIAL ROBUSTNESS

In Section 2, we established that robust and standard models might depend on very different sets of features. We demonstrated how this can lead to a decrease in standard accuracy for robust models. In this section, we will argue that the features learned by robust models can also be beneficial.

At a high level, robustness to adversarial perturbations can be viewed as an invariance property of a model. A model that achieves small loss for all perturbations in the set $\Delta$, will necessarily have learned features that are invariant to such perturbations. Thus, robust training can be viewed as a method to embed certain invariances in a model. Since we also expect humans to be invariant to these perturbations (e.g. small $\ell_p$-bounded changes of the pixels), robust models will be more aligned with human vision than standard models. In this section, we present evidence supporting the view.

**Loss gradients in the input space align well with human perception** As a starting point, we want to investigate which features of the input most strongly affect the prediction of the classifier both for standard and robust models. To this end, we visualize the gradients of the loss with respect to individual features (pixels) of the input in Figure 2. We observe that gradients for adversarially

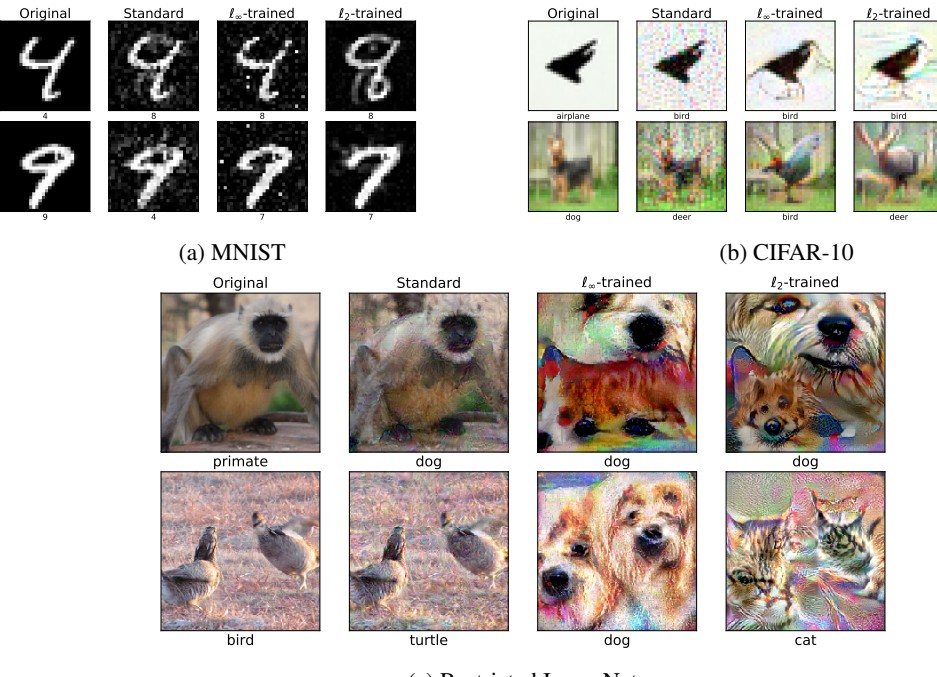

(a) MNIST    (b) CIFAR-10

(c) Restricted ImageNet

Figure 3: Visualizing large-$\varepsilon$ adversarial examples for standard and robust ($\ell_2/\ell_\infty$-adversarial training) models. We construct these examples by iteratively following the (negative) loss gradient while staying with $\ell_2$-distance of $\varepsilon$ from the original image. We observe that the images produced for robust models effectively capture salient data characteristics and appear similar to examples of a different class. (The value of $\varepsilon$ is equal for all models and much larger than the one used for training.) Additional examples are visualized in Figure 8 and 9 of Appendix G.

trained networks align well with perceptually relevant features (such as edges) of the input image. In contrast, for standard networks, these gradients have no coherent patterns and appear very noisy to humans. We want to emphasize that no preprocessing was applied to the gradients (other than scaling and clipping for visualization). On the other hand, extraction of interpretable information from the gradients of standard networks has so far only been possible using additional sophisticated techniques (Simonyan et al., 2013; Yosinski et al., 2015; Olah et al., 2017).

This observation effectively outlines an approach to train models that align better with human perception *by design*. By encoding the correct prior into the set of perturbations $\Delta$, adversarial training alone might be sufficient to yield interpretable gradients. We believe that this phenomenon warrants an in-depth investigation and we view our experiments as only exploratory.

**Adversarial examples exhibit salient data characteristics**    Given how the gradients of standard and robust models are concentrated on qualitatively different input features, we want to investigate how the adversarial examples of these models appear visually. To find adversarial examples, we start from a given test image and apply Projected Gradient Descent (PGD; a standard first-order optimization method) to find the image of highest loss within an $\ell_p$-ball of radius $\varepsilon$ around the original image [2]. This procedure will change the pixels that are most influential for a particular model's predictions and thus hint towards how the model is making its predictions.

The resulting visualizations are presented in Figure 3 (details in Appendix A). Surprisingly, we can observe that adversarial perturbations for robust models tend to produce salient characteristics of another class. In fact, the corresponding adversarial examples for robust models can often be perceived as samples from that class. This behavior is in stark contrast to standard models, for which adversarial examples appear as noisy variants of the input image.

---

[2]To allow for significant image changes, we will use much larger values of $\varepsilon$ than those used during training.

These findings provide additional evidence that adversarial training does not necessarily lead to gradient obfuscation (Athalye et al., 2018). Following the gradient changes the image in a meaningful way and (eventually) leads to images of different classes. Hence, the robustness of these models does not stem from having gradients that are ill-suited for first-order methods.

**Smooth cross-class interpolations via gradient descent**   By linearly interpolating between the original image and the image produced by PGD we can produce a smooth, "perceptually plausible" interpolation between classes (Figure 4). Such interpolation have thus far been restricted to generative models such as GANs (Goodfellow et al., 2014a) and VAEs (Kingma & Welling, 2013), involved manipulation of learned representations (Upchurch et al., 2016), and hand-designed methods (Suwajanakorn et al., 2015; Kemelmacher-Shlizerman, 2016). In fact, we conjecture that the similarity of these inter-class trajectories to GAN interpolations is not a coincidence. We postulate that the saddle point problem that is key in both these approaches may be at the root of this effect. We hope that future research will investigate this connection further and explore how to utilize the loss landscape of robust models as an alternative method to smoothly interpolate between classes.

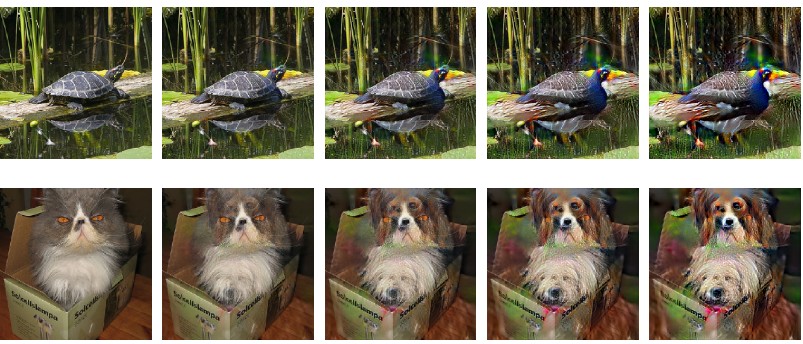

Figure 4: Interpolation between original image and large-$\varepsilon$ adversarial example as in Figure 3.

## 4   RELATED WORK

Due to the large body of related work, we will only focus on the most relevant studies here and defer the full discussion to Appendix F. Fawzi et al. (2018b) prove upper bounds on the robust of classifiers and exhibit a standard vs. robust accuracy trade-off for a specific classifier families on a synthetic task. Their setting also (implicitly) utilizes the notion of robust and non-robust features, however these features have small magnitude rather than weak correlation. Ross & Doshi-Velez (2017) propose regularizing the gradient of the classifier with respect to its input. They find that the resulting classifiers have more interpretable gradients and targeted adversarial examples resemble the target class for digit and character recognition tasks. There has been recent of work proving upper bounds on classifier robustness (Gilmer et al., 2018; Schmidt et al., 2018; Fawzi et al., 2018a). However, this work is orthogonal to ours as in these settings there exist classifiers that are both robust and accurate.

## 5   CONCLUSIONS AND FUTURE DIRECTIONS

In this work, we show that the goal of adversarially robust generalization might fundamentally be at odds with that of standard generalization. Specifically, we identify an inherent trade-off between the standard accuracy and adversarial robustness of a model, that *provably* manifests in a concrete, simple setting. This trade-off stems from intrinsic differences between the feature learned by standard and robust models. Our analysis also explains the drop in standard accuracy observed when employing adversarial training in practice. Moreover, it emphasizes the need to develop robust training methods, since robustness is unlikely to arise as a consequence of standard training.

We discover that even though adversarial robustness comes at a price, it has some unexpected benefits. Robust models learn features that align well with salient data characteristics. The root

of this phenomenon is that the set of adversarial perturbations encodes some prior for human perception. Thus, classifiers that are robust to these perturbations are also necessarily invariant to input modifications that we expect humans to be invariant to. We demonstrate a striking consequence of this phenomenon: robust models yield clean feature interpolations similar to those obtained from generative models such as GANs (Goodfellow et al., 2014b). This emphasizes the possibility of a stronger connection between GANs and adversarial robustness.

Finally, our findings show that the interplay between adversarial robustness and standard classification might be more nuanced that one might expect. This motivates further work to fully undertand the relative costs and benefits of each of these notions.

## ACKNOWLEDGEMENTS

Shibani Santurkar was supported by the National Science Foundation (NSF) under grants IIS-1447786, IIS-1607189, and CCF-1563880, and the Intel Corporation. Dimitris Tsipras was supported in part by the NSF grant CCF-1553428. Aleksander Mądry was supported in part by an Alfred P. Sloan Research Fellowship, a Google Research Award, and the NSF grant CCF-1553428.

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

# A  EXPERIMENTAL SETUP

## A.1  DATASETS

We perform our experimental analysis on the MNIST (LeCun et al., 2010), CIFAR-10 (Krizhevsky & Hinton, 2009) and (restricted) ImageNet (Deng et al., 2009) datasets. For binary classification, we filter out all the images from the MNIST dataset other than the "5" and "7" labelled examples. For the ImageNet dataset, adversarial training is significantly harder since the classification problem is challenging by itself and standard classifiers are already computationally expensive to train. We thus restrict our focus to a smaller subset of the dataset. We group together a subset of existing, semantically similar ImageNet classes into 8 different super-classes, as shown in Table 1. We train and evaluate only on examples corresponding to these classes.

Table 1: Classes used in the Restricted ImageNet model. The class ranges are inclusive.

| Class | Corresponding ImageNet Classes |
| --- | --- |
| "Dog" | 151 to 268 |
| "Cat" | 281 to 285 |
| "Frog" | 30 to 32 |
| "Turtle" | 33 to 37 |
| "Bird" | 80 to 100 |
| "Primate" | 365 to 382 |
| "Fish" | 389 to 397 |
| "Crab" | 118 to 121 |
| "Insect" | 300 to 319 |

## A.2  MODELS

- Binary MNIST (Section 2.2): We train a linear classifier with parameters $w \in \mathbb{R}^{784}, b \in \mathbb{R}$ on the dataset described in Section A.1 (labels $-1$ and $+1$ correspond to images labelled as "5" and "7" respectively). We use the cross-entropy loss and perform 100 epochs of gradient descent in training.

- MNIST: We use the simple convolution architecture from the TensorFlow tutorial (TFM, 2017) [3].
- CIFAR-10: We consider a standard ResNet model (He et al., 2015a). It has 4 groups of residual layers with filter sizes (16, 16, 32, 64) and 5 residual units each [4].
- Restricted ImageNet: We use a ResNet-50 (He et al., 2015a) architecture using the code from the `tensorpack` repository (Wu et al., 2016). We do not modify the model architecture, and change the training procedure only by changing the number of examples per "epoch" from 1,280,000 images to 76,800 images.

### A.3 ADVERSARIAL TRAINING

We perform adversarial training to train robust classifiers following Madry et al. (2017). Specifically, we train against a projected gradient descent (PGD) adversary, starting from a random initial perturbation of the training data. We consider adversarial perturbations in $\ell_p$ norm where $p = \{2, \infty\}$. Unless otherwise specified, we use the values of $\varepsilon$ provided in Table 2 to train/evaluate our models.

Table 2: Value of $\varepsilon$ used for adversarial training/evaluation of each dataset and $\ell_p$-norm.

| Adversary | Binary MNIST | MNIST | CIFAR-10 | Restricted Imagenet |
|---|---|---|---|---|
| $\ell_\infty$ | 0.2 | 0.3 | $4/255$ | 0.005 |
| $\ell_2$ | - | 1.5 | 0.314 | 1 |

### A.4 ADVERSARIAL EXAMPLES FOR LARGE $\varepsilon$

The images we generated for Figure 3 were allowed a much larger perturbation from the original sample in order to produce visible changes to the images. These values are listed in Table 3. Since

Table 3: Value of $\varepsilon$ used for large-$\varepsilon$ adversarial examples of Figure 3.

| Adversary | MNIST | CIFAR-10 | Restricted Imagenet |
|---|---|---|---|
| $\ell_\infty$ | 0.3 | 0.125 | 0.25 |
| $\ell_2$ | 4 | 4.7 | 40 |

these levels of perturbations would allow to truly change the class of the image, training against such strong adversaries would be impossible. Still, we observe that smaller values of $\varepsilon$ suffices to ensure that the models rely on the most robust (and hence interpretable) features.

## B MIXING NATURAL AND ADVERSARIAL EXAMPLES IN EACH BATCH

In order to make sure that the standard accuracy drop in Figure 7 is not an artifact of only training on adversarial examples, we experimented with including unperturbed examples in each training batch, following the recommendation of (Kurakin et al., 2016a). We found that while this slightly improves the standard accuracy of the classifier, it decreases it's robust accuracy by a roughly proportional amount, see Table 4.

## C PROOF OF THEOREM 2.1

The main idea of the proof is that an adversary with $\varepsilon = 2\eta$ is able to change the distribution of features $x_2, \ldots, x_{d+1}$ to reflect a label of $-y$ instead of $y$ by subtracting $\varepsilon y$ from each variable. Hence

---

[3]`https://github.com/MadryLab/mnist_challenge/`
[4]`https://github.com/MadryLab/cifar10_challenge/`

Table 4: Standard and robust accuracy corresponding to robust training with half natural and half adversarial samples. The accuracies correspond to standard, robust and half-half training.

| | Norm | $\varepsilon$ | Standard Accuracy | | | Robust Accuracy | | |
| | | | Standard | Half-half | Robust | Standard | Half-half | Robust |
|---|---|---|---|---|---|---|---|---|
| MNIST | $\ell_\infty$ | 0 | 99.31% | - | - | - | - | - |
| | | 0.1 | 99.31% | 99.43% | 99.36% | 29.45% | 95.29% | 95.05% |
| | | 0.2 | 99.31% | 99.22% | 98.99% | 0.05% | 90.79% | 92.86% |
| | | 0.3 | 99.31% | 99.17% | 97.37% | 0.00% | 89.51% | 89.92% |
| | $\ell_2$ | 0 | 99.31% | - | - | - | - | - |
| | | 0.5 | 99.31% | 99.35% | 99.41% | 94.67% | 97.60% | 97.70% |
| | | 1.5 | 99.31% | 99.29% | 99.24% | 56.42% | 87.71% | 88.59% |
| | | 2.5 | 99.31% | 99.12% | 97.79% | 46.36% | 60.27% | 63.73% |
| CIFAR10 | $\ell_\infty$ | 0 | 92.20% | - | - | - | - | - |
| | | $2/255$ | 92.20% | 90.13% | 89.64% | 0.99% | 69.10% | 69.92% |
| | | $4/255$ | 92.20% | 88.27% | 86.54% | 0.08% | 55.60% | 57.79% |
| | | $8/255$ | 92.20% | 84.72% | 79.57% | 0.00% | 37.56% | 41.93% |
| | $\ell_2$ | 0 | 92.20% | - | - | - | - | - |
| | | $20/255$ | 92.20% | 92.04% | 91.77% | 45.60% | 83.94% | 84.70% |
| | | $80/255$ | 92.20% | 88.95% | 88.38% | 8.80% | 67.29% | 68.69% |
| | | $320/255$ | 92.20% | 81.74% | 75.75% | 3.30% | 34.45% | 39.76% |

any information that is used from these features to achieve better standard accuracy can be used by the adversary to reduce adversarial accuracy. We define $G_+$ to be the distribution of $x_2, \ldots, x_{d+1}$ when $y = +1$ and $G_-$ to be that distribution when $y = -1$. We will consider the setting where $\varepsilon = 2\eta$ and fix the adversary that replaces $x_i$ by $x_i - y\varepsilon$ for each $i \geq 2$. This adversary is able to change $G_+$ to $G_-$ in the adversarial setting and vice-versa.

Consider any classifier $f(x)$ that maps an input $x$ to a class in $\{-1, +1\}$. Let us fix the probability that this classifier predicts class $+1$ for some fixed value of $x_1$ and distribution of $x_2, \ldots, x_{d+1}$. Concretely, we define $p_{ij}$ to be the probability of predicting $+1$ given that the first feature has sign $i$ and the rest of the features are distributed according to $G_j$. Formally,

$$
\begin{aligned}
p_{++} &= \Pr_{x_2,\ldots,d+1 \sim G_+} (f(x) = +1 \mid x_1 = +1), \\
p_{+-} &= \Pr_{x_2,\ldots,d+1 \sim G_-} (f(x) = +1 \mid x_1 = +1), \\
p_{-+} &= \Pr_{x_2,\ldots,d+1 \sim G_+} (f(x) = +1 \mid x_1 = -1), \\
p_{--} &= \Pr_{x_2,\ldots,d+1 \sim G_-} (f(x) = +1 \mid x_1 = -1).
\end{aligned}
$$

Using these definitions, we can express the standard accuracy of the classifier as

$$
\begin{aligned}
\Pr(f(x) = y) &= \Pr(y = +1) \left( p \cdot p_{++} + (1-p) \cdot p_{-+} \right) \\
&\quad + \Pr(y = -1) \left( p \cdot (1 - p_{--}) + (1-p) \cdot (1 - p_{+-}) \right) \\
&= \frac{1}{2} \left( p \cdot p_{++} + (1-p) \cdot p_{-+} + p \cdot (1 - p_{--}) + (1-p) \cdot (1 - p_{+-}) \right) \\
&= \frac{1}{2} \left( p \cdot (1 + p_{++} - p_{--}) + (1-p) \cdot (1 + p_{-+} - p_{+-}) \right).
\end{aligned}
$$

Similarly, we can express the accuracy of this classifier against the adversary that replaces $G_+$ with $G_-$ (and vice-versa) as

$$
\begin{aligned}
\Pr(f(x_{\text{adv}}) = y) &= \Pr(y = +1)\left(p \cdot p_{+-} + (1-p) \cdot p_{--}\right) \\
&\quad + \Pr(y = -1)\left(p \cdot (1 - p_{-+}) + (1-p) \cdot (1 - p_{++})\right) \\
&= \frac{1}{2}\left(p \cdot p_{+-} + (1-p) \cdot p_{--} + p \cdot (1 - p_{-+}) + (1-p) \cdot (1 - p_{++})\right) \\
&= \frac{1}{2}\left(p \cdot (1 + p_{+-} - p_{-+}) + (1-p) \cdot (1 + p_{--} - p_{++})\right).
\end{aligned}
$$

For convenience we will define $a = 1 - p_{++} + p_{--}$ and $b = 1 - p_{-+} + p_{+-}$. Then we can rewrite

$$
\text{standard accuracy}: \quad \frac{1}{2}(p(2 - a) + (1-p)(2 - b))
$$
$$
= 1 - \frac{1}{2}(pa + (1-p)b),
$$
$$
\text{adversarial accuracy}: \quad \frac{1}{2}((1-p)a + pb).
$$

We are assuming that the standard accuracy of the classifier is at least $1 - \delta$ for some small $\delta$. This implies that

$$
1 - \frac{1}{2}(pa + (1-p)b) \geq 1 - \delta \implies pa + (1-p)b \leq 2\delta.
$$

Since $p_{ij}$ are probabilities, we can guarantee that $a \geq 0$. Moreover, since $p \geq 0.5$, we have $p/(1-p) \geq 1$. We use these to upper bound the adversarial accuracy by

$$
\begin{aligned}
\frac{1}{2}((1-p)a + pb) &\leq \frac{1}{2}\left((1-p)\frac{p^2}{(1-p)^2}a + pb\right) \\
&= \frac{p}{2(1-p)}(pa + (1-p)b) \\
&\leq \frac{p}{1-p}\delta.
\end{aligned}
$$

$\square$

## D    PROOF OF THEOREM 2.2

We consider the problem of fitting the distribution $\mathcal{D}$ of (3) by using a standard soft-margin SVM classifier. Specifically, this can be formulated as:

$$
\min_w \mathbb{E}\left[\max(0, 1 - yw^\top x)\right] + \frac{1}{2}\lambda\|w\|_2^2 \tag{5}
$$

for some value of $\lambda$. We will assume that we tune $\lambda$ such that the optimal solution $w^*$ has $\ell_2$-norm of 1. This is without much loss of generality since our proofs can be adapted to the general case. We will refer to the first term of (5) as the *margin* term and the second term as the *regularization* term.

First we will argue that, due to symmetry, the optimal solution will assign equal weight to all the features $x_i$ for $i = 2, \ldots, d+1$.

**Lemma D.1.** *Consider an optimal solution $w^*$ to the optimization problem* (5). *Then,*
$$
w_i^* = w_j^* \ \forall \, i, j \in \{2, ..., d+1\}.
$$

*Proof.* Assume that $\exists \, i, j \in \{2, ..., d+1\}$ such that $w_i^* \neq w_j^*$. Since the distribution of $x_i$ and $x_j$ are identical, we can swap the value of $w_i$ and $w_j$, to get an alternative set of parameters $\hat{w}$ that has the same loss function value ($\hat{w}_j = w_i$, $\hat{w}_i = w_j$, $\hat{w}_k = w_k$ for $k \neq i, j$).

Moreover, since the margin term of the loss is convex in $w$, using Jensen's inequality, we get that averaging $w^*$ and $\hat{w}$ will not increase the value of that margin term. Note, however, that $\|\frac{w^* + \hat{w}}{2}\|_2 < \|w^*\|_2$, hence the regularization loss is strictly smaller for the average point. This contradicts the optimality of $w^*$. $\square$

Since every optimal solution will assign equal weight to all $x_i$ for $k \geq 2$, we can replace these features by their sum (and divide by $\sqrt{d}$ for convenience). We will define

$$z = \frac{1}{\sqrt{d}} \sum_{i=2}^{d+1} x_i,$$

which, by the properties of the normal distribution, is distributed as

$$z \sim \mathcal{N}(y\eta\sqrt{d}, 1).$$

By assigning a weight of $v$ to that combined feature the optimal solutions can be parametrized as

$$w^\top x = w_1 x_1 + vz,$$

where the regularization term of the loss is $\lambda(w_1^2 + v^2)/2$.

Recall that our chosen value of $\eta$ is $4/\sqrt{d}$, which implies that the contribution of $vz$ is distributed normally with mean $4yv$ and variance $v^2$. By the concentration of the normal distribution, the probability of $vz$ being larger than $v$ is large. We will use this fact to show that the optimal classifier will assign on $v$ at least as much weight as it assigns on $w_1$.

**Lemma D.2.** *Consider the optimal solution $(w_1^*, v^*)$ of the problem* (5). *Then*

$$v^* \geq \frac{1}{\sqrt{2}}.$$

*Proof.* Assume for the sake of contradiction that $v^* < 1/\sqrt{2}$. Then, with probability at least $1 - p$, the first feature predicts the wrong label and without enough weight, the remaining features cannot compensate for it. Concretely,

$$\mathbb{E}[\max(0, 1 - yw^\top x)] \geq (1 - p) \, \mathbb{E}\left[\max\left(0, 1 + w_1 - \mathcal{N}\left(4v, v^2\right)\right)\right]$$
$$\geq (1 - p) \, \mathbb{E}\left[\max\left(0, 1 + \frac{1}{\sqrt{2}} - \mathcal{N}\left(\frac{4}{\sqrt{2}}, \frac{1}{2}\right)\right)\right]$$
$$> (1 - p) \cdot 0.016.$$

We will now show that a solution that assigns zero weight on the first feature ($v = 1$ and $w_1 = 0$), achieves a better margin loss.

$$\mathbb{E}[\max(0, 1 - yw^\top x)] = \mathbb{E}\left[\max\left(0, 1 - \mathcal{N}\left(4, 1\right)\right)\right]$$
$$< 0.0004.$$

Hence, as long as $p \leq 0.975$, this solution has a smaller margin loss than the original solution. Since both solutions have the same norm, the solution that assigns weight only on $v$ is better than the original solution $(w_1^*, v^*)$, contradicting its optimality. $\qquad\square$

We have established that the learned classifier will assign more weight to $v$ than $w_1$. Since $z$ will be at least $y$ with large probability, we will show that the behavior of the classifier depends entirely on $z$.

**Lemma D.3.** *The standard accuracy of the soft-margin SVM learned for problem* (5) *is at least* 99%.

*Proof.* By Lemma D.2, the classifier predicts the sign of $w_1 x_1 + vz$ where $vz \sim \mathcal{N}(4yv, v^2)$ and $v \geq 1/\sqrt{2}$. Hence with probability at least 99%, $vzy > 1/\sqrt{2} \geq w_1$ and thus the predicted class is $y$ (the correct class) independent of $x_1$. $\qquad\square$

We can utilize the same argument to show that an adversary that changes the distribution of $z$ has essentially full control over the classifier prediction.

**Lemma D.4.** *The adversarial accuracy of the soft-margin SVM learned for* (5) *is at most 1% against an $\ell_\infty$-bounded adversary of $\varepsilon = 2\eta$.*

*Proof.* Observe that the adversary can shift each feature $x_i$ towards $y$ by $2\eta$. This will cause $z$ to be distributed as

$$z_{\text{adv}} \sim \mathcal{N}(-y\eta\sqrt{d}, 1).$$

Therefore with probability at least $99\%$, $vyz < -y \leq -w_1$ and the predicted class will be $-y$ (wrong class) independent of $x_1$. $\qquad\square$

It remains to show that adversarial training for this classification task with $\varepsilon > 2\eta$ will results in a classifier that has relies solely on the first feature.

**Lemma D.5.** *Minimizing the adversarial variant of the loss (5) results in a classifier that assigns $0$ weight to features $x_i$ for $i \geq 2$.*

*Proof.* The optimization problem that adversarial training solves is

$$\min_w \max_{\|\delta\|_\infty \leq \varepsilon} \mathbb{E}\big[\max(0, 1 - yw^\top(x + \delta))\big] + \frac{1}{2}\lambda\|w\|_2^2,$$

which is equivalent to

$$\min_w \mathbb{E}\big[\max(0, 1 - yw^\top x + \varepsilon\|w\|_1)\big] + \frac{1}{2}\lambda\|w\|_2^2.$$

Consider any optimal solution $w$ for which $w_i > 0$ for some $i > 2$. The contribution of terms depending on $w_i$ to $1 - yw^\top x + \varepsilon\|w\|_1$ is a normally-distributed random variable with mean $2\eta - \varepsilon \leq 0$. Since the mean is non-positive, setting $w_i$ to zero can only decrease the margin term of the loss. At the same time, setting $w_i$ to zero *strictly* decreases the regularization term, contradicting the optimality of $w$. $\qquad\square$

Clearly, such a classifier will have standard and adversarial accuracy of $p$ against any $\varepsilon < 1$ since such a value of $\varepsilon$ is not sufficient to change the sign of the first feature. This concludes the proof of the theorem.

# E  ROBUSTNESS-ACCURACY TRADE-OFF: AN EMPIRICAL EXAMINATION

Our theoretical analysis shows that there is an inherent tension between standard accuracy and adversarial robustness. At the core of this trade-off is the concept of robust and non-robust features. The robustness of a feature is characterized by the strength of its correlation with the correct label. It is natural to wonder whether this concept of robust features is an artifact of our theoretical analysis or if it manifests more broadly. We thus investigate this issue experimentally on a dataset that is amenable to linear classifiers, MNIST (LeCun et al., 1998) (details in Appendix A).

Recall the goal of standard classification for linear classifiers is to predict accurately, i.e. $y = \text{sign}(w^\top x)$. Hence the correlation of a feature $i$ with the true label, computed as $|\mathbb{E}[yx_i]|$, quantifies how useful this feature is for classification. In the adversarial setting, against an $\varepsilon$ $\ell_\infty$-bounded adversary we need to ensure that $y = \text{sign}(w^\top x - \varepsilon y\|w\|_1)$. In that case we expect a feature $i$ to be helpful if $|\mathbb{E}[yx_i]| \geq \varepsilon$.

This calculation suggests that in the adversarial setting, there is an implicit threshold on feature correlations imposed by the threat model (the perturbation allowed to the adversary). While standard models may utilize all features with non-zero correlations, a robust model cannot rely on features with correlation below this threshold. In Figure 5(b), we visualize the correlation of each pixel (feature) in the MNIST dataset along with the learned weights of the standard and robust classifiers. As expected, we see that the standard classifier assigns weights even to weakly-correlated pixels so as to maximize prediction confidence. On the other hand, the robust classifier does not assign any weight below a certain correlation threshold which is dictated by the adversary's strength ($\varepsilon$) (Figures 5(a, b))

Interestingly, the standard model assigns non-zero weight even to very weakly correlated pixels (Figure 5(a)). In settings with finite training data, such non-robust features could arise from noise. (For instance, in $N$ tosses of an unbiased coin, the expected imbalance between heads and tails is $O(\sqrt{N})$ with high probability.) A standard classifier would try to take advantage of even this "hallucinated" information by assigning non-zero weights to these features.

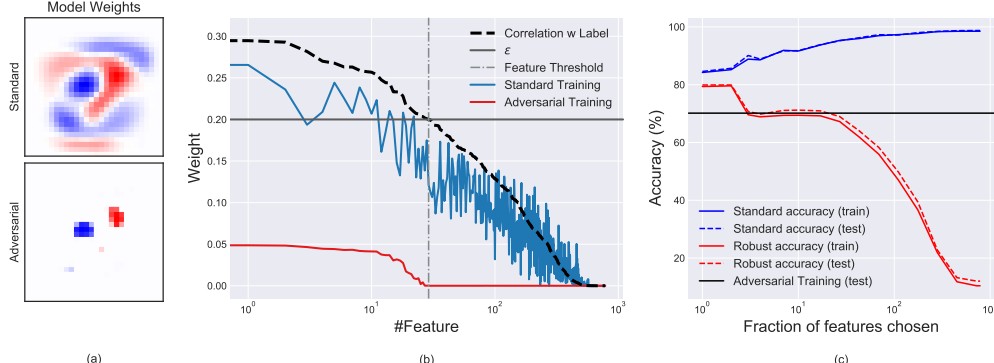

(a)  (b)  (c)

Figure 5: Analysis of linear classifier trained on a binary MNIST task (5 vs. 7). (Details in Appendix Table 5.) (a) Visualization of network weights per input feature. (b) Comparison of feature-label correlation to the weight assigned to the feature by each network. Adversarially trained networks put weights only on a small number of strongly-correlated or "robust" features. (c) Performance of a model trained using *standard* training only on the most robust features. Specifically, we sort features based on decreasing correlation with the label and train using only the most correlated ones. Beyond a certain threshold, we observe that as more non-robust or (weakly correlated) features are available to the model, the standard accuracy increases at the cost of robustness.

### E.1 AN ALTERNATIVE PATH TO ROBUSTNESS?

The analysis above highlights an interesting trade-off between the predictive power of a feature and its vulnerability to adversarial perturbations. This brings forth the question – Could we use these insights to train robust classifiers with *standard* methods (i.e. without performing adversarial training)? As a first step, we train a (standard) linear classifier on MNIST utilizing input features (pixels) that lie above a given correlation threshold (see Figure 5(c)). As expected, as more non robust features are incorporated in training, the standard accuracy increases at the cost of robustness. Further, we observe that a standard classifier trained in this manner using few robust features attains better robustness than even adversarial training. This results suggest a more direct (and potentially better) method of training robust networks in certain settings.

## F ADDITIONAL RELATED WORK

Fawzi et al. (2016) derive parameter-dependent bounds on the robustness of any fixed classifier. Our results focus on the statistical setting itself and provide lower bounds for *all* classifiers learned in this setting.

Wang et al. (2017) analyze the adversarial robustness of nearest neighbor classifiers. Instead we focus on lower bounds that are inherent to the statistical setting itself and apply to all classifiers.

Schmidt et al. (2018) study the generalization aspect of adversarially robustness. They show that the number of samples needed to achieve adversarially robust generalization is polynomially larger in the dimension than the number of samples needed to ensure standard generalization. However, in the limit of infinite data, one can learn classifiers that are both robust and accurate.

Gilmer et al. (2018) demonstrate a setting where even a small amount of standard error implies that most points provably have a misclassified point close to them. In this setting, achieving perfect standard accuracy (easily achieved by a simple classifier) is sufficient to achieve perfect adversarial robustness. In contrast, our work focuses on a setting where adversarial training (provably) matters and there exists a trade-off between standard and adversarial accuracy.

Xu & Mannor (2012) explore the connection between robustness and generalization, showing that, in a certain sense, robustness can imply generalization. This direction is orthogonal to our, since we work in the limit of infinite data, optimizing the distributional loss directly.

Fawzi et al. (2018a) prove lower bounds on the robustness of any classifier based on certain generative assumptions. Since these bounds apply to all classifiers, independent of architecture and training procedure, they fail to capture the situation we face in practice where robust optimization can significantly improve the adversarial robustness of standard classifiers (Madry et al., 2017; Kolter & Wong, 2017; Raghunathan et al., 2018; Sinha et al., 2017).

A recent work (Bubeck et al., 2018) turns out to (implicitly) rely on the distinction between robust and non-robust features in constructing a distribution for which adversarial robustness is hard from a different, computational point of view.

Goodfellow et al. (2014b) observed that adversarial training results in feature weights that depend on fewer input features (similar to Figure 5(a)). Additionally, it has been observed that for *naturally trained* RBF classifiers on MNIST, targeted adversarial attacks resemble images of the target class (Goodfellow, 2015).

Su et al. (2018) empirically observe a similar trade-off between the accuracy and robustness of *standard* models across different deep architectures on ImageNet.

Babbar & Schölkopf (2018) study an extreme multi-label problem and observe that for classes with relatively few examples, $\ell_1$-regularization (which corresponds to adversarial training for linear models) is helpful, while for classes with more samples, it is harmful to the model accuracy.

## G OMITTED FIGURES

Table 5: Comparison of performance of linear classifiers trained on a binary MNIST dataset with standard and adversarial training. The performance of both models is evaluated in terms of standard and adversarial accuracy. Adversarial accuracy refers to the percentage of examples that are correctly classified after being perturbed by the adversary. Here, we use an $\ell_\infty$ threat model with $\varepsilon = 0.20$ (with images scaled to have coordinates in the range $[0,1]$).

| | Standard Accuracy (%) | | Adversarial Accuracy (%) | | $\|w\|_1$ |
| --- | --- | --- | --- | --- | --- |
| | Train | Test | Train | Test | |
| Standard Training | 98.38 | 92.10 | 13.69 | 14.95 | 41.08 |
| Adversarial Training | 94.05 | 70.05 | 76.05 | 74.65 | 13.69 |

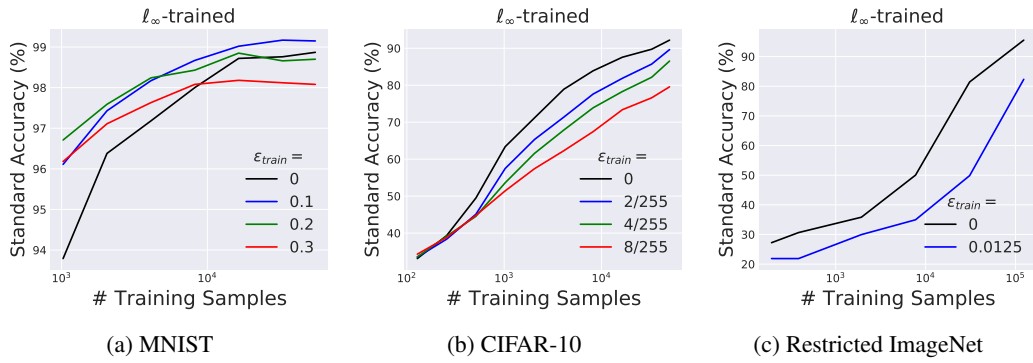

(a) MNIST      (b) CIFAR-10      (c) Restricted ImageNet

Figure 6: Comparison of standard accuracies of models trained against an $\ell_\infty$-bounded adversary as a function of the size of the training dataset. We observe that in the low-data regime, adversarial training has an effect similar to data augmentation and helps with generalization in certain cases (particularly on MNIST). However, in the limit of sufficient training data, we see that the standard accuracy of robust models is less than that of the standard model ($\varepsilon_{train} = 0$), which supports the theoretical analysis in Section 2.1.

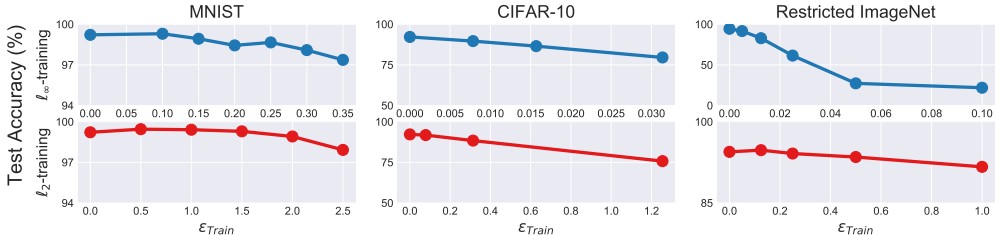

Figure 7: Standard test accuracy of adversarially trained classifiers. The adversary used during training is constrained within some $\ell_p$-ball of radius $\varepsilon_{\text{train}}$ (details in Appendix A). We observe a consistent decrease in accuracy as the strength of the adversary increases.

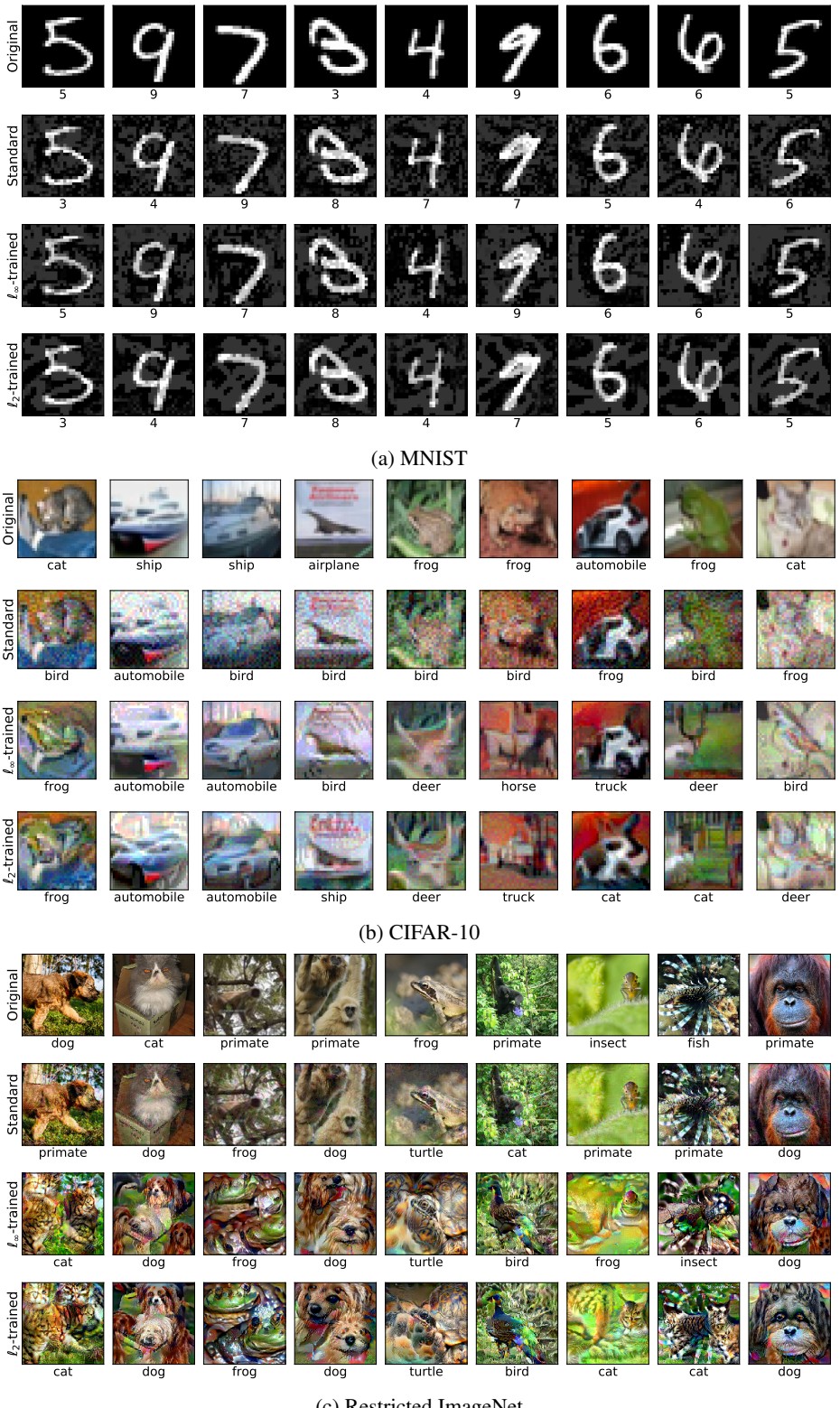

Figure 8: Large-$\varepsilon$ adversarial examples, bounded in $\ell_\infty$-norm, similar to those in Figure 3.

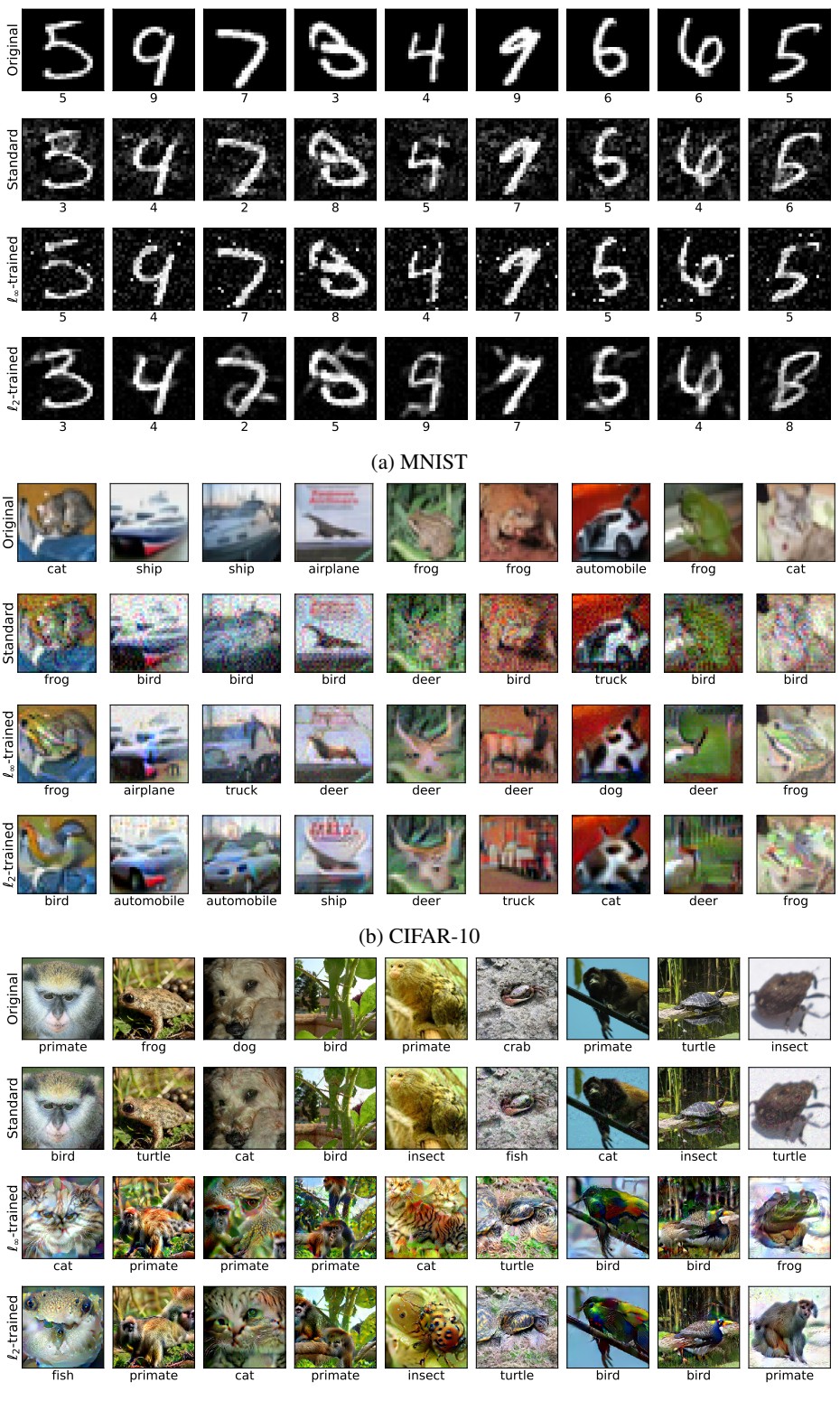

Figure 9: Large-$\varepsilon$ adversarial examples, bounded in $\ell_2$-norm, similar to those in Figure 3.

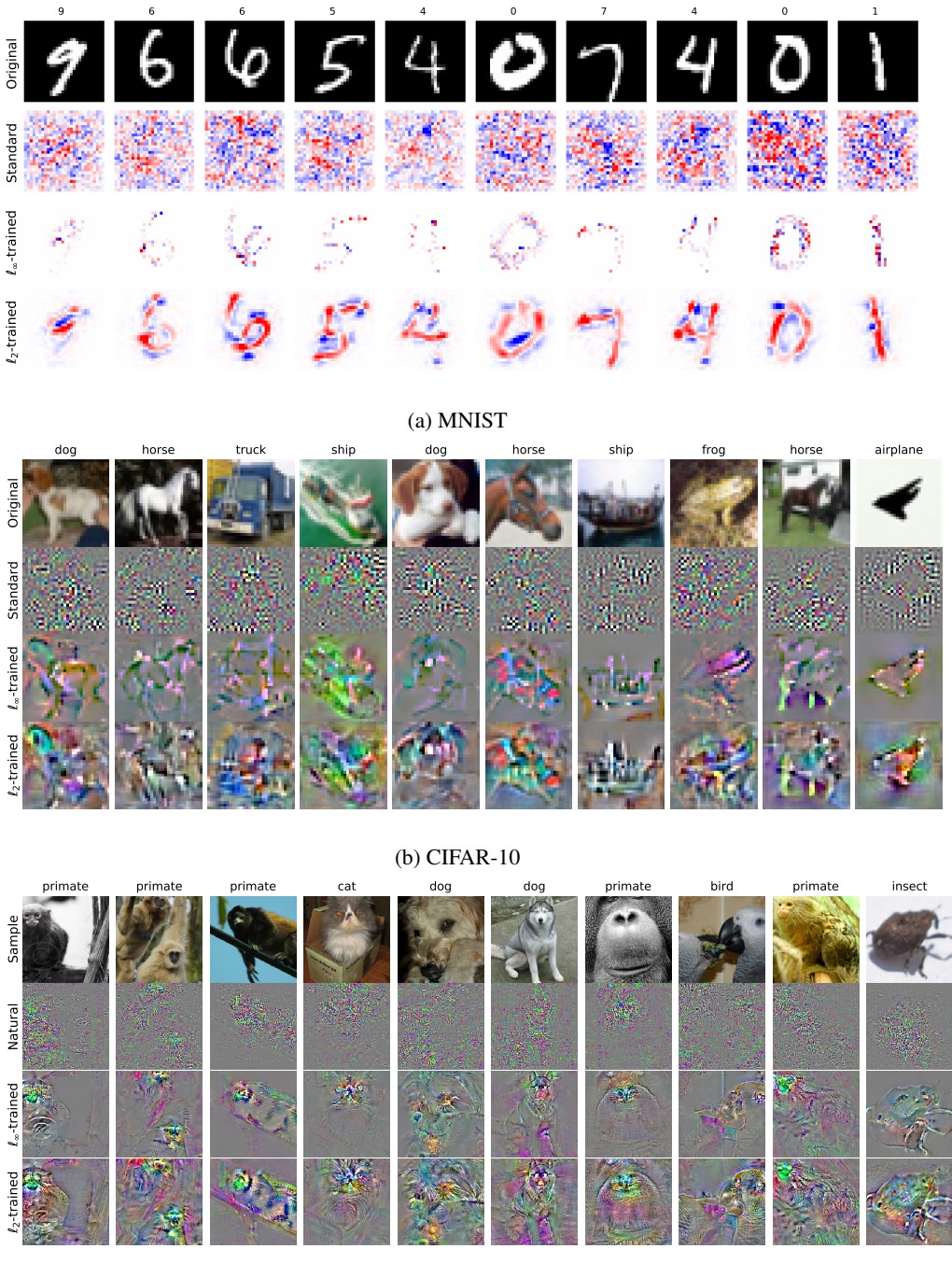

Figure 10: Visualization of the gradient of the loss with respect to input features (pixels) for standard and adversarially trained networks for 10 randomly chosen samples, similar to those in Figure 2. Gradients are significantly more *interpretable* for adversarially trained networks – they align almost perfectly with perceptually relevant features. For MNIST, blue and red pixels denote positive and negative gradient regions respectively. For CIFAR10 and Restricted ImageNet we clip pixel to 3 standard deviations and scale to $[0, 1]$.

