# OpenReview forum: "Robustness May Be at Odds with Accuracy"
_ICLR.cc/2019/Conference_

### Official Review · AnonReviewer2 · 2018-11-02
**Good paper, clear accept**

**Rating:** 8
**Confidence:** 2

**Review:**

The paper demonstrates the trade-off between accuracy and robustness of a model. The phenomenon is shown in previous works, but this work interestingly proposes a theoretical model that supports the idea. The proving technique can be particularly beneficial to developing theoretical understanding for the phenomenon. Besides, the authors also visualize the gradients and adversarial examples generated from standard and adversarially trained models, which show that these adversarially trained models are more aligned to human perception.

Quality: good, clarity: good, originality: good, significance: good

Pros:
- The paper is fairly well written and the idea is clearly presented
- To the best of my knowledge (maye I am wrong), this work is the first one that
provides theoretical explanation for the tradeoff between accuracy and robustness
- The visualization results supports their hypothesis that adversarially trained models
percepts more like human.

Suggestions:
It would be interesting to see what kind of real images can fool the models and see whether the robust model made mistakes more like human.

---

> ### Author Response · Authors · 2018-11-14
> **Author response**
>
> We thank the reviewer for their kind comments. The reviewer’s suggestion about the nature of errors made by standard vs. robust models is really interesting, and we will pursue it in future work.

---

### Official Review · AnonReviewer1 · 2018-11-02
**good paper, interesting findings, should be cautious on over-claiming**

**Rating:** 7
**Confidence:** 4

**Review:**

This paper discusses the hypothesis of the existence of intrinsic tradeoffs between clean accuracy and robust accuracy and corresponding implications. Specifically, it is motivated by the tradeoffs between clean accuracy and robust accuracy of adversarially trained network. The authors constructed a toy example and proved that any classifier cannot be both accurate and robust at the same time. They also showed that regular training cannot make soft-margin SVM robust but adversarial training can. At the end of the paper, they show that input gradients of adversarially trained models are more semantically meaningful than regularly trained models.

The paper is well written and easy to follow. The toy example is novel and provides a concrete example demonstrating robustness-accuracy tradeoff, which was previously speculated. Demonstrating adversarially trained models has more semantically meaningful gradient is interesting and provides insights to the field. It connects robustness and interpretability nicely.

My main concern is on the overclaiming of applicability of the "inherent tradeoff". The paper demonstrated that the "inherent tradeoff" could be a reasonable hypothesis for explaining the difficulty of achieving robust models. I think the authors should emphasize this in the paper so that it does not mislead the reader to think that it is the reason.

On a related note, Theorem 2.2 shows adversarial training can give robust classifier while standard training cannot. Then the paper says "adversarial training is necessary to achieve non-trivial adversarial accuracy in this setting". The word "necessary" is misleading, here Thm 2.2 showed that adversarial training works, but it doesn't exclude the possibility that robust classifiers can be achieved by other training methods.

minor comments
- techinques --> techniques
- more discussion on the visual difference between the gradients from L2 and L_\infty adversarially trained networks
- Figure 5 (c): what does "w Robust Features" mean? are these values accuracy after perburtation?

---

> ### Author Response · Authors · 2018-11-14
> **Author response**
>
> We thank the reviewer for the kind comments and suggestion. We address concerns raised below:
>
> - We agree with the reviewer that "inherent trade-off" might be perceived incorrectly. We only intended to refer to an inherent tradeoff in *our setting*. While we do argue that this is a reasonable hypothesis for the difficulties we face in practice, we cannot definitively conclude that this is the case. We have edited the manuscript to reflect this.
>
> - We agree that alternative methods can be used to obtain robustness in Thm 2.2. We only stated that "adversarial training is necessary" because we wanted to emphasize that simply minimizing the standard loss (ignoring the adversary) will not lead to robustness. We have edited the manuscript to elaborate on this.
>
> We thank the reviewer for the other comments. We have edited the manuscript to address them.

---

### Official Review · AnonReviewer3 · 2018-11-04
**interesting findings, however seems to confirm some of the already known behavior in linear classification setup**

**Rating:** 8
**Confidence:** 3

**Review:**

This paper presents a study of tradeoffs between adversarial and standard accuracy of classifiers. Though it might be expected that training for adversarial robustness always leads to improvement in standard accuracy, however the authors claim that the actual situation is quite subtle. Though adversarial training might help towards increasing standard accuracy in certain data regimes such as data scarcity, but when sufficient data is available there exists a trade-off between the two goals. The tradeoff is demonstrated in a fairly simple setting in which case data consists of two kinds of features - those which are weakly correlated with the output, and those which are strongly correlated. It is shown that adversarial accuracy depends on the feature which exhibit strong correlation, while standard accuracy depends on weakly correlated features.

Though the paper presents some interesting insights. Some of the concerns  are :
 - The paper falls short in answering the tradeoff question under a more general setup. The toy example is very specific with a clear separation between weak and strongly correlated features. It would be interesting to see how similar results can be derived when under more complicated setup with many features with varying extent of correlation.
 - The tradeoff between standard and robustness under linear classification has also been demonstrated in a recent work [1]. In [1], it is also argued that for datasets consisting of large number of labels, when some of the labels are under data-scarce regimes, an adversarial robustness view-point (via l1-regularization) helps in accuracy improvement for those labels. However, for other set of labels for which there is sufficient data available,  l2-regularization is more suited, and adversarial robustness perspective decreases standard accuracy. From this view-point, one could argue that some of the main contributions in the current paper, could be seen as empirical extensions for deep learning setup. It would be instructive to contrast and explore connections between this paper, and the observations in [1].
[1] Adversarial Extreme Multi-label Classification, https://arxiv.org/abs/1803.01570
==============post-rebuttal======
thanks for the feedback, I update my rating of the paper

---

> ### Author Response · Authors · 2018-11-14
> **Author response**
>
> We thank the reviewer for the detailed comments. We will address the concerns raised below:
>
> - The aim of our paper is to demonstrate an inherent trade off between robustness and standard accuracy in a concrete setting. We believe that exhibiting the tradeoff in a simple and natural setting is a strength rather than a weakness of our paper, since such simple settings can manifest as special cases of more complex settings. We want to emphasize that our proof does not depend on the specific setting in any crucial way. In particular, the proof can be straightforwardly extended to a more general setting where each feature is an independent Gaussian with a different mean (and thus different correlation with the label).
>
> The main idea is that, for a given adversary in this setting, we can always separate the features into "robust" (utilizing these features can only help robust classification) and "non-robust" (the adversary can manipulate these features to a degree where they become harmful for the model's accuracy). Any feature with correlation less than a threshold determined by epsilon is considered as non-robust in this context. Hence, a robust classifier cannot rely on these non-robust features.
>
> As a result, if there is any standard accuracy that can be gained by utilizing these non-robust features, the model trained in standard way will benefit from it (at the expense of reducing its robust accuracy) and the robust model will not be able to get such a benefit, leading to its standard accuracy being lower.
>
> Thus the trade-off discussed in the paper would manifest as long as there are some non-robust features which contribute to the accuracy of the standard model. Since extending our results to such settings would be fairly routine, we decided to keep our setting simple and highlight the key principle at play.
>
> - We thank the reviewer for bringing this paper to our attention. We added a discussion of the paper in the related work discussion. We want to emphasize that our goal is to understand and theoretically demonstrate the standard vs. robust accuracy tradeoffs observed in practice (reported multiple times in prior work as we discuss in our paper, as well as in the suggested paper). We are not claiming to be the first ones to observe tradeoffs of this nature _empirically_, but we are the first to provide some insight into its roots.

---

### Public Comment · (anonymous) · 2018-11-22
**The effect of adversarial training on standard accuracy**

Nice paper! You have provided the empirical results on how the adversarial training hurts the standard accuracy in the high data regime. While in Theorem 2.1, you proved how the robust accuracy can be upper bounded for a given standard accuracy, there is no proof of how the standard accuracy is upper bounded for a given robust accuracy.

Is that right? or I am missing something?

---

> ### Author Response · Authors · 2018-11-22
> **Author response**
>
> Thank you for your interest in our paper. In Theorem 2.1 we are proving upper bounds on the *robust accuracy* for a given *standard accuracy* (e.g. standard accuracy >95% implies robust accuracy <45%). One can consider the contrapositive to obtain bounds on the *standard accuracy* for a given *robust accuracy*. That is "If the robust accuracy is at least p * δ / (1-p) then the standard accuracy has to be <1 - δ" (i.e. any classifier with at least 45% robust accuracy cannot have standard accuracy more than 95%).

---

### Meta-Review · Area_Chair1 · 2018-12-13
**Good paper. Accept.**

**Confidence:** 5
**Recommendation:** Accept (Poster)

**Metareview:**

This paper provides interesting discussions on the trade-off between model accuracy and robustness to adversarial examples. All reviewers found that both empirical studies and theoretical results are solid. The paper is very well written. The visualization results are very intuitive. I recommend acceptance.